# Locally Valid and Discriminative Prediction Intervals for Deep Learning Models

**Zhen Lin**
University of Illinois at Urbana-Champaign
Urbana, IL 61801
zhenlin4@illinois.edu

**Shubhendu Trivedi**
MIT
Cambridge, MA 02139
shubhendu@csail.mit.edu

**Jimeng Sun**
University of Illinois at Urbana-Champaign
Urbana, IL 61801
jimeng@illinois.edu

## Abstract

Crucial for building trust in deep learning models for critical real-world applications is efficient and theoretically sound uncertainty quantification, a task that continues to be challenging. Useful uncertainty information is expected to have two key properties: It should be *valid* (guaranteeing coverage) and *discriminative* (more uncertain when the expected risk is high). Moreover, when combined with deep learning (DL) methods, it should be *scalable* and *affect the DL model performance minimally*. Most existing Bayesian methods lack frequentist coverage guarantees and usually affect model performance. The few available frequentist methods are rarely discriminative and/or violate coverage guarantees due to unrealistic assumptions. Moreover, many methods are expensive or require substantial modifications to the base neural network. Building upon recent advances in conformal prediction [13, 33] and leveraging the classical idea of kernel regression, we propose Locally Valid and Discriminative prediction intervals (LVD), a simple, efficient and lightweight method to construct discriminative prediction intervals (PIs) for almost *any* DL model. With no assumptions on the data distribution, such PIs also offer finite-sample local coverage guarantees (contrasted to the simpler marginal coverage). We empirically verify, using diverse datasets, that besides being the only locally valid method for DL, LVD also exceeds or matches the performance (including coverage rate and prediction accuracy) of existing uncertainty quantification methods, while offering additional benefits in scalability and flexibility.

## 1  Introduction

Consider a training set $\mathcal{S}_{\text{train}} = \{(X_i, Y_i)\}_{i=1}^N$ and a test example $(X_{N+1}, Y_{N+1})$, all drawn i.i.d from an arbitrary joint distribution $\mathcal{P}$, with $(X_i, Y_i) \in \mathcal{X} \times \mathcal{Y}$ for some $\mathcal{X} \subseteq \mathbb{R}^d$ and $\mathcal{Y} \subseteq \mathbb{R}$. We are interested in the problem of predictive inference: On observing $\mathcal{S}_{\text{train}}$ and $X_{N+1}$, our task is to construct a prediction interval (PI) [1] estimate $\hat{C}(X_{N+1})$ that contains the true value of $Y_{N+1}$ with a (pre-specified) high probability.

---

[1]Several recent deep learning papers use "Confidence Interval" and "Prediction Interval" interchangeably. We stick to the conventional statistical usage.

35th Conference on Neural Information Processing Systems (NeurIPS 2021).

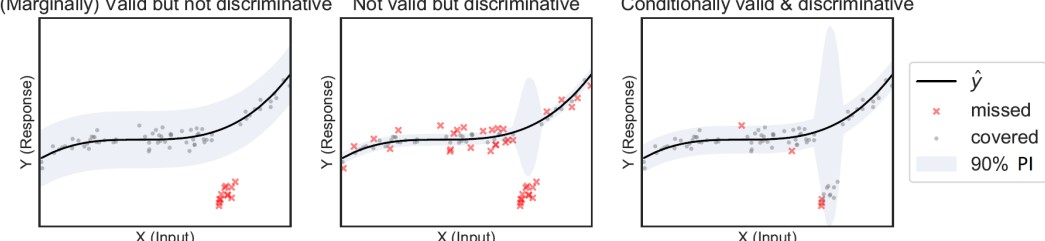

Figure 1: Illustration of possible (good and bad) PIs. The PI on the left is valid, as it covers 90% of the data. It is however only marginally valid, not reflecting the poor model prediction near the red cluster. The middle PI is discriminative, and reflects the high error near the red cluster, but its coverage rate is much lower than the target (thus not valid). The PI on the right addresses both challenges by stretching "just enough" near the cluster, making it not only discriminative, but also *conditionally* valid. We seek to construct PIs of the last type, but constructing exact conditionally valid PIs in a distribution-free setting is theoretically impossible. We thus relax this goal by instead aiming for local validity (more details in Section 2).

The construction of actionable PIs involves two *general challenges*: First, $\hat{C}$ should be **valid**, meaning that if the specified probability is $1-\alpha$, we expect $\hat{C}(X_{N+1})$ to cover $Y_{N+1}$ at least $1-\alpha$ of the time. Moreover, $\hat{C}$ should be **discriminative** i.e., we expect $\hat{C}(X_{N+1})$ to be narrower for confident cases and vice-versa. The width of the PI $\hat{C}(X_{N+1})$ is thus quantification of the uncertainty. Figure 1 illustrates these notions, with more details in Section 2.2 and 2.3.

While deep learning (DL) models have demonstrated impressive performance over a range of complicated tasks and data modalities, it has remained difficult to quantify the uncertainty for their predictions. For DL predictions to be actionable, uncertainty information is however indispensable, especially in domains like medicine and finance [2]. Apart from requiring validity and discrimination as discussed earlier, two additional challenges exist *specifically* for DL models. Obviously, any uncertainty estimation method needs to finish reasonably fast to be useful, so the third challenge is *scalability*. The fourth challenge is *accuracy*: The uncertainty estimation should not decrease the prediction accuracy of the DL model. *Post-hoc* methods are ideal because they usually do not interfere with the base NN prediction at all. These four requirements together constitute a set of essential desiderata for uncertainty quantification in DL.

Existing uncertainty estimation methods for DL rarely address more than one or two of the above requirements. Credible intervals given by posteriors of approximate Bayesian methods such as [43, 15], deep ensemble [20, 44] and Monte-Carlo dropout [12] are not valid in the frequentist sense [7]. Most existing methods also interfere with the original model design, loss function and/or training, which could be expensive and decrease the model performance (as verified in our experiments) [12, 1, 7, 20].

To address these requirements, we leverage recent advances in conformal prediction and the classical idea of kernel regression. Conformal prediction, pioneered by Vovk [40], is a powerful approach for constructing valid PIs. The most popular split conformal methods usually leverage prediction errors from a hold-out set to construct $\hat{C}(X_{N+1})$, which would be valid if a future data point $(X_{N+1}, Y_{N+1})$ follows the same distribution as data in the hold-out set. This framework is particularly suitable for deep learning due to its distribution-free nature, and has motivated many recent uncertainty quantification efforts in deep learning for both classification and regression tasks [1, 23, 3, 11]. However, most conformal methods are only marginally valid [25, 40, 21, 4]. Moreover, less-than-meticulous applications to DL can break distributional assumptions and theoretical validity, as in the case of [1] (see discussion in Appendix). We however seek to construct a PI conditioning on the input (similar to the third PI in Fig. 1). Some recent advances ([13, 33]) examine the possibility of "approximately" conditionally valid PIs. While these methods cannot be directly applied to DL due to efficiency and performance considerations, their methodological and theoretical contributions serve as major inspirations for us to develop a highly flexible and practical method.

**Summary of Contributions**: We propose Locally Valid Discriminative Prediction Intervals (LVD), a simple uncertainty estimation method for deep learning which combines recent advances in conformal prediction and the classical idea of kernel regression. LVD applies to almost all DL models, and is the first method that satisfies all four aforementioned requirements:

• Validity: LVD has frequentist coverage guarantee (not just marginal, but approximately conditional).

- Discrimination: The width of the PIs given by LVD adapts to the risk/uncertainty level of $X_{N+1}$.
- Scalability: LVD is lightweight, adding limited overhead to the base DL model.
- Accuracy: LVD is post-hoc without requiring model retraining, and does not affect the base performance of the DL model.

We must note that while the theoretical foundation for guaranteeing "validity" is mostly based on [13], LVD addresses several challenges to satisfy the other three requirements. The code to replicate all our results can be found at `https://github.com/zlin7/LVD`.

## 2 Preliminaries

### 2.1 Learning Setup and Assumptions

We assume data and response pairs $(X, Y) \in \mathcal{X} \times \mathcal{Y}$ have a joint distribution denoted $\mathcal{P}$, with the marginal distributions of $Y$ and $X$ and the conditional distribution $Y|X$ denoted as $\mathcal{P}_Y$, $\mathcal{P}_X$, and $\mathcal{P}_{Y|X}$, respectively. Further, we will define $Z_i := (X_i, Y_i)$ for concision.

Assuming that we already have an algorithm (with all the training protocols folded in), such as a Deep Neural Network (DNN), that provides a mean estimator $\hat{\mu}^{NN}(x) : \mathcal{X} \mapsto \mathcal{Y}$. Given a target coverage level $1 - \alpha \in (0, 1)$, our task is to also construct a prediction interval estimator function $\hat{C}_\alpha(x) : \mathcal{X} \mapsto \{\text{subset of } \mathcal{Y}\}$ that has the **validity** and **discrimination** properties as defined below.

### 2.2 Validity (Frequentist Coverage)

There are several (related) notions for a PI to be valid – marginal, conditional, and local. Given target level $1 - \alpha$, we say $\hat{C}_\alpha$ has the **marginal coverage** guarantee (or, equivalently, is marginally valid) if

$$\mathbb{P}\{Y_{N+1} \in \hat{C}_\alpha(X_{N+1})\} \geq 1 - \alpha \tag{1}$$

where the probability is taken over the training data and (the unseen) $(X_{N+1}, Y_{N+1})$.

A limitation of marginal coverage is that it is not conditioned on $X_{N+1}$. A more desirable, albeit stronger, property would be **conditional coverage** at $1 - \alpha$:

$$\mathbb{P}\{Y_{N+1} \in \hat{C}_\alpha(X_{N+1})|X_{N+1} = x\} \geq 1 - \alpha \text{ for almost all } x \in \mathcal{X}. \tag{2}$$

Here the probability is taken over the training data and $Y_{N+1}$ (with $X_{N+1}$ fixed). It is thus clear that conditional coverage implies marginal coverage but not the other way around. Indeed, a $\hat{C}_\alpha$ with marginal coverage property only implies a $1 - \alpha$ chance of being accurate *on average* across all data points (marginalizing over $X_{N+1}$) i.e. there might be a sub-population in the data for which the coverage is completely missed. Unfortunately, it is *impossible* to achieve distribution-free finite-sample conditional coverage (Eq. 2) in a non-trivial way. Indeed, it is known that a finite-sample estimated $\hat{C}_\alpha(x)$ cannot achieve conditional coverage, unless it produces infinitely wide prediction intervals in expectation under any non-discrete distribution $\mathcal{P}$ [39, 22, 5].

It is thus reasonable to instead seek *approximate* conditional coverage. As might be apparent, there is considerable freedom in defining an appropriate notion of "approximate", depending on specific tasks and domains. However, a sufficiently general-purpose and natural notion involves using a kernel function $K : \mathcal{X} \times \mathcal{X} \mapsto \mathbb{R}$ and a center $x' \in \mathcal{X}$, like the relaxation given in [33]:

$$\frac{\int \mathbb{P}\{Y_{N+1} \in \hat{C}_\alpha(x')|X_{N+1} = x\} K(x, x') d\mathcal{P}_X(x)}{\int K(x, x') d\mathcal{P}_X(x)} \geq 1 - \alpha \tag{3}$$

with the probability ($\mathbb{P}$ in the integral) taken over all training samples and $Y_{N+1}$, with $(X_{N+1}, Y_{N+1}) \sim \tilde{\mathcal{P}} = \tilde{\mathcal{P}}_X \times \mathcal{P}_{Y|X}$. Here $\tilde{\mathcal{P}}_X$ is just the distribution re-weighted by the kernel with a center $x'$, defined by $\frac{d\tilde{\mathcal{P}}_X(x)}{dx} \propto \frac{d\mathcal{P}_X(x)}{dx} K(x', x)$. Instead of choosing $x'$ beforehand, if we let the center be $X_{N+1}$ and fold the integral into $\mathbb{P}$ like in [13], we arrive at the definition of **local coverage**:

$$\mathbb{P}\{\tilde{Y}_{N+1} \in \hat{C}_\alpha(X_{N+1})|X_{N+1} = x'\} \geq 1 - \alpha. \tag{4}$$

Here the probability integrates over all training data and an additional $(\tilde{X}_{N+1}, \tilde{Y}_{N+1}) \sim \tilde{\mathcal{P}}$ defined above. Intuitively, this definition means $\hat{C}_\alpha(X_{N+1})$ is valid "on average" within a small neighborhood

of $X_{N+1}$. Note that Eqs. 1 and 2 reduce to Eq. 4 with $K$ being constant and delta functions, respectively. In the rest of the paper, we will call $\hat{C}_\alpha$ marginally/conditionally/locally valid if it satisfies Eq. 1/2/4 respectively, and we will pursue finite-sample **local validity**.

## 2.3 Discrimination

The idea of discrimination is simple: If the error of our prediction $\hat{\mu}(x)$ is high for an input $x$, the PI should be wide, and vice versa. Formally, following [1], we require

$$\mathbb{E}[W(\hat{C}(x))] \geq \mathbb{E}[W(\hat{C}(x'))] \Leftrightarrow \mathbb{E}[\ell(y, \hat{\mu}(x))] \geq \mathbb{E}[\ell(y', \hat{\mu}(x'))]. \tag{5}$$

Here the expectation is taken over the training data, $W$ is a measure of the width of the PI, and $\ell$ is a loss function such as MSE. This property can be verified (as shown in Section 4) by checking how well $W(\hat{C}(x))$ could predict the magnitude of the error. Discrimination could be considered a measure of efficiency, as a good $\hat{C}$ could "save" some width when the expected risk is low. However, it only makes sense to compare efficiency if all else is equal (i.e. two marginally valid PIs estimators with the same error). Note that although discrimination could be related to conditional/local validity, they are not the same - e.g., a PI that is always infinitely wide is conditionally valid, but not discriminative.

Our goal is to achieve both local validity and discrimination without making any assumptions about the underlying distribution $\mathcal{P}$ (i.e., in a distribution-free setting). As noted in Section 1, our method should also run fast and not affect the performance of underlying neural network model $\hat{\mu}^{NN}$.

## 3 Method: Locally Valid Discriminative Prediction Intervals (LVD)

**Overview**: We first train a deep neural network (DNN) $\hat{\mu}^{NN}$ (if not already given), followed by a post-hoc training of an appropriately chosen kernel function $K$. Specifically, we learn $K$ in a non-parametric kernel regression setting using embeddings from the deep learning model while optimizing for the underlying distance metric that the kernel function leverages. Both of these steps are explicated in more detail in Section 3.1. Armed with $\hat{\mu}^{NN}$, we proceed to utilize a hold-out set to collect prediction residuals, which are used with the learned $K$ (along with its distance metric) to build the final PI for any datum at inference time (Section 3.2). We then show the finite-sample local validity and asymptotic conditional validity in Section 3.4.

### 3.1 Training

At the onset, we partition $\mathcal{S}_{\text{train}}$ of $N$ data points into two sets - $\mathcal{S}_{\text{embed}}$ and $\mathcal{S}_{\text{conformal}}$. We will denote $\mathcal{S}_{\text{embed}}$ as $\{Z_i\}_{i=1}^n$ and $\mathcal{S}_{\text{conformal}}$ as $\{Z_{n+i}\}_{i=1}^m$, where $m = N - n$. $\mathcal{S}_{\text{embed}}$ is used to learn an embedding function $\mathbf{f}$ and a kernel $K$, and $\mathcal{S}_{\text{conformal}}$ is used for conformal prediction.

**[Optional] Training an Embedding Function**: Instead of training a deep kernel in a kernel regression directly, which can be prohibitively expensive, we split the training task into two steps: training the (expensive) DNN, and training the kernel $K$. Specifically, we first train a DNN mean estimator $\hat{\mu}^{NN} : \mathcal{X} \mapsto \mathcal{Y}$ to solve the supervised regression task with the mean squared error (MSE) loss. Note that $\hat{\mu}^{NN}$ can be based on any existing model. Moreover, this step could be skipped if we are already provided with a pre-trained $\hat{\mu}^{NN}$. Then, we remove the last layer of $\hat{\mu}^{NN}$ and produce an embedding function $\mathbf{f} : \mathcal{X} \mapsto \mathbb{R}^h$ for some positive integer $h$. If the original model $\hat{\mu}^{NN}$ is good, usually such an embedding provides a rich and discriminative representation of the input (as will be verified empirically in Section 4).

**Training the Kernel**: Fixing the embedding funtion $\mathbf{f}$, we perform leave-one-out Nadaraya-Watson [24][14][41] kernel regression with a learnable Gaussian kernel on $\mathcal{S}_{\text{embed}}$:

$$\hat{y}_i^{KR} = \frac{\sum_{j \neq i, j \in [n]} y_j K_{\mathbf{f}}(x_i, x_j)}{\sum_{j \neq i, j \in [n]} K_{\mathbf{f}}(x_i, x_j)} \tag{6}$$

$$\text{where } K_{\mathbf{f}}(x_i, x_j) = K(\mathbf{f}(x_i), \mathbf{f}(x_j)) = \frac{1}{\sigma\sqrt{2\pi}} e^{\frac{-d(\mathbf{f}(x_i), \mathbf{f}(x_j))}{\sigma^2}} \text{ and } [n] := \{1, \dots, n\}. \tag{7}$$

Here $d(\cdot, \cdot)$ is a Mahalanobis distance parameterized by a positive-semidefinite matrix $\mathbf{W} \succeq 0$, which is learned. To avoid solving an expensive semi-definite program, instead of working with $\mathbf{W}$ directly,

we work with a low-rank matrix $\mathbf{A} \in \mathbb{R}^{h \times k}$ such that $\mathbf{W} = \mathbf{A}^T \mathbf{A}$, yielding the following equivalent distance formulation:

$$d(\mathbf{f}(x_i), \mathbf{f}(x_j)) = \|\mathbf{A}(\mathbf{f}(x_i) - \mathbf{f}(x_j))\|^2. \tag{8}$$

This parameterization of $K$ is similar to that in [42]. Finally, to train $K$, we minimize the MSE loss.

**Residual Collection**: In this step, we take the trained embedding function and kernel, denoted as $K_{\mathbf{f}}$ for simplicity, and apply it on $\mathcal{S}_{\text{conformal}}$. $\forall i \in [m]$, we compute and collect the absolute residual,

$$R_i = |y_{n+i} - \hat{y}_{n+i}|, \tag{9}$$

the distribution of which is used for PI construction. It is important to remark that $\hat{y}$ does not have to be $\hat{y}^{KR}$. The main purpose of the previous step is to train the $K$, and $\hat{y}$ could still be obtained through the original DNN $\hat{y} = \hat{\mu}^{NN}(x)$, or *any* estimator not trained on $\mathcal{S}_{\text{conformal}}$. As a result, the accuracy can only improve (if $\hat{y}^{KR}$ turns out to be a better mean estimator)[2].

## 3.2 Inference

Before proceeding further, we recall a useful definition and fix some necessary notation. For a distribution with cumulative density function (cdf) $F$ defined on the augmented real line $\mathbb{R} \cup \{-\infty, \infty\}$, the quantile function is defined as $Q(\alpha, F) = F^{-1}(\alpha)$. This definition is the same for a finite distribution like the empirical distribution. Suppose the empirical distribution consists of $R_1, \ldots, R_m$, then we denote the empirical distribution $\hat{F}$ and the empirical quantile $Q(\alpha, \hat{F})$ as:

$$\hat{F} = \frac{1}{m} \sum_{i=1}^{m} \delta_{R_i} \qquad \text{and} \qquad Q(\alpha, \hat{F}) = \inf_r \hat{F}(r) \geq \alpha \tag{10}$$

where $\delta_R(r) = \mathbb{1}\{r \geq R\}$. Note that we treat $\{R_i\}_{i=1}^m$ as an unordered list. Besides, $R_i$ can be $\pm\infty$, and can repeat. Finally, we can assign weights to $R_i$, and define the quantiles for a weighted distribution:

$$\tilde{F} = \sum_{i=1}^{m} w_i \delta_{R_i} \qquad \text{where} \qquad \sum_{i=1}^{m} w_i = 1. \tag{11}$$

**Split Conformal**: Before presenting the detailed construction of the PI in LVD, it would be particularly instructive to first consider a special case. Specifically, when $K$ returns a constant number for any $(x_i, x_j)$, we recover the well-known "split conformal" method [25, 40, 21], which uses the $1 - \alpha$ quantile of the residuals as the PI width. Following our setup, the split conformal PI is given by:

$$\hat{C}_\alpha^{split}(X_{N+1}) = \left\{ y \in \mathbb{R} : |y - \hat{y}_{N+1}| \leq Q\left(1 - \alpha, \frac{1}{m+1}\left(\delta_\infty + \sum_{i=1}^{m} \delta_{R_i}\right)\right) \right\}. \tag{12}$$

Because the residuals $\{R_i\}_{i \in [m]} \cup \{R_{N+1}\}$ are i.i.d., $R_{N+1}$'s ranking among them is uniformly distributed. We cannot know $R_{N+1}$, so we use $\infty$ instead to be "safe" ($\forall r \in \mathbb{R}, \delta_\infty(r) = 0$). It follows that $\hat{C}_\alpha^{split}$ is $(1 - \alpha)$ marginally valid [25].

**Local Conformal**: In order to achieve the local coverage property, all we need to do is to re-weigh the residuals. Following the approach in [13], we arrive at the following suitable notion of PI:

$$\hat{C}_\alpha^{LVD}(X_{N+1}) = \left\{ y \in \mathbb{R} : |y - \hat{y}_{N+1}| \leq Q\left(1 - \alpha, \left(w_{N+1}\delta_\infty + \sum_{i=1}^{m} w_{n+i}\delta_{R_i}\right)\right) \right\} \tag{13}$$

$$\text{where} \quad w_j = \frac{K_{\mathbf{f}}(x_j, x_{N+1})}{K_{\mathbf{f}}(x_{N+1}, x_{N+1}) + \sum_{i=1}^{m} K_{\mathbf{f}}(x_{n+i}, x_{N+1})}. \tag{14}$$

In other words, we first assign weights to $\{R_i\}$ based on the similarity between $\{X_{n+i}\}$ and $X_{N+1}$ using $K_{\mathbf{f}}$, and then set the width to be the weighted quantile . Note that with $\delta_\infty$, $\hat{C}_\alpha^{LVD}(X_{N+1})$ will be infinitely wide if data is scarce around $X_{N+1}$. However, as argued in [13], this is desired.

---

[2]As will be shown in the Appendix, $\hat{y}^{KR}$ is often preferable because of the distance information it encodes.

### 3.3 Implementation Details

**Parameterization and Training**: Since $\mathbf{A}$ is intricately linked to the computation of the weights assigned by the Gaussian kernel $K$ (Eq. 8), it is implemented as $K(\mathbf{f}(x_i), \mathbf{f}(x_j)) = e^{-||\mathbf{A}(\mathbf{f}(x_i) - \mathbf{f}(x_j))||^2}$. In order to optimize for $\mathbf{A}$, we treat it as a usual linear layer in a neural network and perform gradient descent.

**Smoothness Requirement**: In the context of obtaining locally valid prediction intervals, a potential drawback of using the Nadaraya-Watson kernel regression framework is that the $K_{\mathbf{f}}$ will not meaningfully learn the similarity of any input with *itself*. For example, we can arbitrarily define $K_{\mathbf{f}}(x, x)$ to be *any* value, including $\infty$. In the context of only regression, the fitted function's performance will not change as long as there are no two identical $x_i$. With the Gaussian kernel, this issue is somewhat mitigated. However, during the training, the $K(x_i, x_i)$ can still be too high compared with $K(x_i, x_i + \epsilon)$, resulting in a less meaningful definition for local coverage. We could then enforce a regularization by replacing the $\hat{y}_i$ in Eq. 6 with

$$\hat{y}_i'^{KR} = \frac{\overline{y}_{-i} K_{\mathbf{f}}(x_i, x_i) + \sum_{j \neq i, j \in [n]} y_j K_{\mathbf{f}}(x_i, x_j)}{K_{\mathbf{f}}(x_i, x_i) + \sum_{j \neq i, j \in [n]} K_{\mathbf{f}}(x_i, x_j)} \quad \text{where } \overline{y}_{-i} = \frac{1}{n-1} \sum_{j \neq i, j \in [n]} y_j. \quad (15)$$

This can be considered an explicit bias term towards the (leave-one-out) sample mean. Empirically, we observe that enforcing this requirement is crucial to obtain meaningful and tight intervals. We direct the reader to the Appendix for a detailed ablation on its utility.

**Complexity**: To facilitate training, we use stochastic gradient descent instead of gradient descent with batch size denoted as $B_1$. Furthermore, if the dataset size is prohibitively large, we can also randomly sample a subset of $B_2$ points $\{x_j\}_{j \neq i}$ to predict $\hat{y}_i$. The total complexity is $O(B_1 B_2 hk)$ where $h$ and $k$, defined earlier, denote the dimensionality of the embedding before/after it is multiplied by $\mathbf{A}$. Note that $B_2 = O(1)$ or $o(N)$. The inference time for each data point can be improved from $O(B_2 hk)$ to $O(B_2 k + hk)$ by storing $\mathbf{A} x_j$ instead of $\mathbf{A}$ and $x_j$ separately.

Denoting the number of parameter of the base NN as $P$, since DL models are usually overparameterized, the additional training time for each descent could be comparable or shorter than training the base NN (depending on the relation between $P$ and $B_2 hk$)[3], and the additional inference time would be much shorter than that of the base NN model. In addition, most of these factors (especially $B_2$) can be easily parallelized. The full procedure is summarized below in Algorithm 1.

---

**Algorithm 1** LVD

**Input**:
$\mathcal{S}_{\text{train}}$: A set of observations $\{Z_i = (X_i, Y_i)\}_{i=1}^{N}$
$\alpha$: Parameter specifying (local) target coverage rate
$X_{N+1}$: Unseen data point
**Output**: A locally valid PI, $\hat{C}_\alpha(X_{N+1})$.
**Training**:
    [Optional] Randomly split $\mathcal{S}_{\text{train}}$ into $\mathcal{S}_{\text{embed}}$ and $\mathcal{S}_{\text{conformal}}$. Denote $\mathcal{S}_{\text{conformal}}$ as $\{Z_{n+i}\}_{i=1}^{m}$
    [Optional] Train a NN regression model $\hat{\mu}^{NN}$ on $\mathcal{S}_{\text{embed}}$.
    Remove the last layer of $\hat{\mu}^{NN}$ to get an embedding function $\mathbf{f}$
    Train $\mathbf{A}$ on $\mathbf{f}(\mathcal{S}_{\text{embed}})$ in a Nadaraya-Watson kernel regression setting, with kernel $K_{\mathbf{f}}(x_1, x_2) = e^{-||\mathbf{A}(\mathbf{f}(x_1) - \mathbf{f}(x_2))||^2}$
    Collect residuals $R_i = |y_{n+i} - \hat{y}_{n+i}|$ for $i \in [m]$
**Inference**:

Compute PI as $\hat{C}_\alpha(X_{N+1}) = \left\{ y \in \mathbb{R} : |y - \hat{y}_{N+1}| \leq Q\left(1 - \alpha, w_{N+1}\delta_\infty + \sum_{i=1}^{m} w_{n+i}\delta_{R_i}\right) \right\}$

where $w_j = \dfrac{K_{\mathbf{f}}(x_j, x_{N+1})}{K_{\mathbf{f}}(x_{N+1}, x_{N+1}) + \sum_{i=1}^{m} K_{\mathbf{f}}(x_{n+i}, x_{N+1})}$

---

[3] In practice, since $\mathbf{f}$ is already well-trained, the training of $K_{\mathbf{f}}$ converges very fast.

## 3.4 Theoretical Guarantees

We conclude this section by showing that $\hat{C}_\alpha^{LVD}$ provides the local coverage property. We adapt Theorem 5.1 in [13] and results in [33] to our setting. The detailed proof is deferred to the Appendix:

**Theorem 3.1.** *Conditional on $X_{N+1}$, the PI obtained from Algorithm 1, $\hat{C}_\alpha^{LVD}(X_{N+1})$, satisfies*

$$\mathbb{P}\{\tilde{Y}_{N+1} \in \hat{C}_\alpha^{LVD}(X_{N+1})|X_{N+1} = x'\} \geq 1 - \alpha \text{ for any } x' \tag{16}$$

*where the probability is taken over all the training samples $\overset{i.i.d.}{\sim} \mathcal{P} = \mathcal{P}_{Y|X} \times \mathcal{P}_X$, and $(\tilde{X}_{N+1}, \tilde{Y}_{N+1})$ with distribution $\tilde{X}_{N+1}|X_{N+1} \sim \mathcal{P}_X^{X_{N+1}}$ and $\tilde{Y}_{N+1}|\tilde{X}_{N+1} \sim \mathcal{P}_{Y|X}$. Here $\mathcal{P}_X^{X_{N+1}}$ means the localized distribution with $\frac{d\mathcal{P}_X^{X_{N+1}}(x)}{dx} \propto \frac{d\mathcal{P}_X(x)}{dx} K_{\mathbf{f}}(X_{N+1}, x)$.*

With some regularity assumptions like in [22], we can also obtain asymptotic conditional coverage:

**Theorem 3.2.** *With appropriate assumptions, $\hat{C}^{LVD}$ is asymptotically conditional valid.*

The detailed assumptions, formal statement, and proof of Theorem 3.2 are deferred to the Appendix.

**Remark:** Roughly speaking, Theorem 3.1 tells us that the response $Y$ of a new data point sampled "near" $X_{N+1}$ will fall in our PI with high probability. Theorem 3.2 further states that, under suitable assumptions and enough data, $\hat{C}^{LVD}$ also covers $Y_{N+1}$ (i.e., no re-sampling) with high probability.

# 4 Experiments

**Baselines**: We compare LVD with the following baselines (with a qualitative comparison in Table 1):

1. *Discriminative Jackknife (DJ)* [1], which claims to be both discriminative and marginally valid but is neither (See Appendix C).
2. *Deep Ensemble (DE)* [20], which trains an ensemble of networks to estimate variance and mean.
3. *Monte-Carlo Dropout (MCDP)* [12], a popular bayesian method for NN that performs Dropout [31] at inference time for the predictive variance estimate.
4. *Probabilistic Backpropagation (PBP)* [15], a successful method to train Bayesian Neural Networks by computing a forward propagation of probabilities before a backward computation of gradients.
5. *Conforamlized Quantile Regression (CQR)* [29], an efficient (narrow PI) marginally valid conformal method that takes *quantile* predictors instead of mean predictors. This comes with a huge cost: one needs to retrain the predictor for *each* $\alpha$ if more than one coverage level is desired.
6. *MAD-Normalized Split Conformal (MADSplit)* [21, 8], a variant of the well-known split-conformal method that requires an estimator for the mean absolute deviation (MAD), and performs conformal prediction on the MAD-normalized residuals.

In our experiments, PIs for non-valid methods are obtained from the quantile functions of the posterior for target coverage $1 - \alpha$ like in [1].

## 4.1 Synthetic Data

We will first examine the dynamics of different uncertainty methods with synthetic data. The formula we use is the same as in [15, 1]: $y = x^3 + \epsilon$. Here, $\epsilon \sim \mathcal{N}(0, 4^2)$, and $x$ comes from $Unif[-1, 1]$ with probability 0.9, and half-normal distribution on $[1, \infty)$ with $\sigma = 1$ with probability 0.1. We used this $\mathcal{P}_X$ to illustrate local validity. The results are shown in Figure 2. We observe that LVD, CQR, MADSplit, and DJ all achieve close to 90% coverage. However, LVD gives a more meaningful discriminative predictive band: Specifically, near the boundaries, it will give us wider intervals (sometimes $\infty$) because there is little similar data around, which is desirable for local validity. Although CQR and MADSplit can be discriminative, they are still only marginally valid, so we can see that despite the varying width, they actually get narrower when $x$ is more eccentric, which is clearly an issue. DJ essentially gives PIs of constant width, as estimated from the quantile of the residuals. DE also does not give meaningful uncertainty estimates, giving almost constant PIs that

Table 1: Features of different methods. CQR and MADSplit achieve strict finite-sample *marginal* coverage. Unlike LVD, no baseline is locally valid. LVD, MADSplit and CQR are discriminative, DJ is not, and DE/CMDP/PBP are supposed to be but usually fail to in our experiment. Among the post-hoc methods, LVD and MADSplit have reasonable overhead, DJ's overhead is usually $O(N)$ where $N$ is the number of training data (and extremely large memory consumption). CQR is not post-hoc because its PI may not contain $\hat{\mu}^{NN}(X)$.

|  | LVD | MADSplit | CQR | DJ | DE | MCDP | PBP |
|---|---|---|---|---|---|---|---|
| Valid | Local | Marginal | Marginal | no guarantee | × | × | × |
| Discriminative | ✓ | ✓ | ✓ | × | sometimes | sometimes | sometimes |
| Post-hoc | ✓ | ✓ | × | ✓ | × | × | × |
| Overhead (if post-hoc) | Low | Low | N/A | Very High | N/A | N/A | N/A |

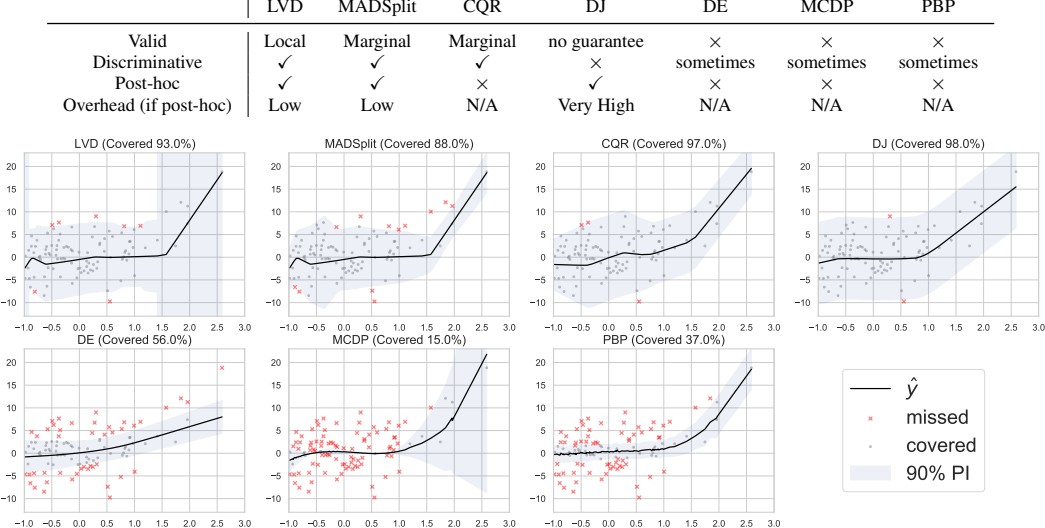

Figure 2: LVD, CQR and MADSplit achieve marginal coverage. Although without theoretical guarantee, DJ usually marginally covers in practice with near constant width (not discriminative). DE, MCDP and PBP do not show validity as expected. Among valid PIs, only LVD tries to capture the less representative points by wider PIs, arguably showing the most useful discriminative pattern.

cover well below 90%. For Bayesian methods, MCDP behaves much like a Gaussian Process (as claimed in [12]), with low coverage rate, whereas PBP is mildly discriminative and not valid[4].

## 4.2 Real Datasets

Table 2: Size of each dataset. Size of the test set is in parenthesis.

| Yacht | Housing | Energy | Bike | Kin8nm | Concrete | QM8 | QM9 |
|---|---|---|---|---|---|---|---|
| 308 (62) | 506 (101) | 768 (154) | 17379(3476) | 8192 (1638) | 1030 (206) | 21786 (4357) | 133719 (26744) |

We will be using a series of standard benchmark datasets in the uncertainty literature [1, 29, 15], including: UCI Yacht Hydrodynamics (Yacht) [38], UCI Bikesharing (Bike) [35], UCI Energy Efficiency (Energy) [37], UCI Concrete Compressive Strength (Concrete) [36], Boston Housing (Housing) [9], Kin8nm [16].We also use QM8 (16 sub-tasks) and QM9 (12 sub-tasks) [28, 30, 27] as examples of more complicated datasets. In each experiment, 20% of the data is used for testing. The sizes of datasets used are shown in Table 2. We use the same DNN model for all baselines, which has 2 layers, 100 hidden nodes each layer, and ReLU activation for the non-QM datasets. For QM8 and QM9, we use the molecule model implemented in [45] and apply applicable baselines. Missing baselines ("–" in the tables) are either too expensive (i.e. time and/or memory) or require a significant redesign of the training and NN, which is beyond the scope of this paper.

**Evaluation Metrics**: The evaluation is based on validity and discrimination. For validity, we check the marginal coverage rate (MCR) and the tail coverage rate (TCR), which is defined as the coverage rate for data whose $Y$ falls in the top and bottom 10%. The motivation behind TCR is that if our local validity is very close to conditional validity, then LVD's coverage rate would be above target in *any* pre-defined sub-samples, including those with extreme $Y$s. For discrimination, to verify Eq. 5, which

---

[4]Sometimes it may be possible to calibrate Bayesian methods [29]. However, one needs to calibrate the entire posterior for the Bayesian method to makes sense. Moreover, from our experiments, it is impossible to do this in MCDP, when it behaves like a Gaussian process and predicts zero variance near known data.

is a prediction task, we compute the AUROC of using the PI width to predict whether the absolute residual is in the top half of all residuals. AUROC alone is misleading, however, as a bad predictor can easily be discriminative (e.g., by randomly adding to both its prediction and PI width a huge constant). Therefore, we also report the mean absolute deviation (MAD), defined as $\frac{\sum_{i=1}^{M} |\hat{y}_i - y_i|}{M}$.

Table 3: Marginal coverage rate (MCR) and tail coverage rate (TCR) (coverage rate for left and right 10% tail for test label) with target at 90%. "–" represents not-applicable models (see Section 4.2). Coverage rates not significantly lower than target at $p = 0.05$ are in bold (good). Note that the too high is not better. For example, MCDP either *greatly* over- or under-covers with MCR either 100% or well below 90%.

| MCR | LVD | MADSplit | CQR | DJ | DE | MCDP | PBP |
|---|---|---|---|---|---|---|---|
| Yacht | **96.8**±2.2 | 82.4±7.1 | **91.5**±4.7 | **95.0**±2.1 | 22.7±6.0 | 87.4±4.2 | 80.2±10.8 |
| Housing | **96.8**±2.9 | **90.6**±3.5 | **91.7**±3.3 | **97.6**±1.4 | **96.0**±1.9 | 100.0±0.0 | 8.1±4.3 |
| Energy | **94.0**±1.6 | **90.3**±2.5 | **90.3**±2.2 | **96.2**±1.8 | 98.0±2.7 | 100.0±0.0 | 7.2±5.9 |
| Bike | **90.4**±0.8 | **89.9**±0.6 | **89.8**±0.7 | **95.2**±0.6 | 100.0±0.0 | 71.9±0.7 | 0.6±0.2 |
| Kin8nm | **98.0**±0.6 | **90.0**±0.8 | **90.2**±0.6 | **94.7**±0.4 | 100.0±0.1 | 100.0±0.0 | 100.0±0.0 |
| Concrete | **97.4**±1.3 | 88.8±2.8 | 88.5±2.3 | **98.0**±1.6 | **97.8**±1.0 | 100.0±0.0 | 3.3±0.8 |
| QM8* | **92.6**±0.9 | **90.0**±0.7 | **90.0**±0.6 | – | 100.0±0.0 | – | – |
| QM9* | **90.3**±0.6 | **90.0**±0.2 | **90.0**±0.3 | – | 60.7±46.8 | – | – |

| TCR | LVD | MADSplit | CQR | DJ | DE | MCDP | PBP |
|---|---|---|---|---|---|---|---|
| Yacht | **98.5**±3.2 | 65.4±23.8 | 77.7±12.3 | 76.2±9.9 | 1.5±4.9 | 50.0±9.8 | 70.0±14.3 |
| Housing | **96.2**±4.4 | **87.6**±8.8 | 82.9±8.2 | **90.0**±5.2 | 81.9±9.7 | 100.0±0.0 | 1.0±3.0 |
| Energy | **86.8**±5.8 | 78.4±10.9 | 73.5±12.0 | **90.0**±6.5 | **95.8**±6.3 | 100.0±0.0 | 9.7±12.7 |
| Bike | **90.2**±1.7 | **89.2**±3.5 | 58.7±7.3 | 85.6±3.3 | 100.0±0.0 | 49.9±0.0 | 0.0±0.0 |
| Kin8nm | **97.2**±1.6 | 86.4±2.6 | 85.2±2.2 | 88.1±1.8 | 99.9±0.3 | 100.0±0.0 | 100.0±0.0 |
| Concrete | **97.1**±3.4 | 83.9±7.3 | 85.4±6.2 | **95.6**±3.6 | **91.7**±4.8 | 100.0±0.0 | 3.4±5.7 |
| QM8* | **90.8**±1.9 | 86.3±2.4 | 80.0±5.9 | – | 100.0±0.0 | – | – |
| QM9* | **89.7**±2.5 | 86.1±3.0 | 79.7±8.9 | – | 60.3±46.5 | – | – |

Table 4: At $p = 0.05$, AUROCs (in predicting error being greater than 50% percentile) that are significantly higher than 50%, and mean absolute deviations (MAD) significantly lower than the second-best, are in bold. LVD, MADSplit and CQR are consistently discriminative, but CQR sometimes incurs high MAD. DJ is not discriminative, whereas other methods occasionally demonstrate discrimination but usually have high MADs as well.

| AUROC | LVD | MADSplit | CQR | DJ | DE | MCDP | PBP |
|---|---|---|---|---|---|---|---|
| Yacht | **83.5**±5.8 | **77.7**±9.0 | **84.9**±4.6 | 50.0±10.5 | **59.8**±6.4 | 47.2±7.7 | **82.8**±8.8 |
| Housing | **59.2**±8.5 | **62.0**±8.3 | **62.5**±6.7 | 49.6±5.9 | **60.0**±6.8 | 42.5±7.8 | 47.4±3.6 |
| Energy | **73.5**±6.3 | **72.9**±5.6 | **72.1**±8.2 | 57.5±8.1 | 56.1±11.0 | **54.6**±5.5 | 48.2±2.6 |
| Bike | **68.2**±11.0 | **71.7**±8.5 | **84.8**±33.5 | 45.8±6.2 | **86.2**±12.5 | **94.3**±1.0 | 48.3±1.0 |
| Kin8nm | **60.3**±1.1 | **60.4**±1.9 | **60.0**±2.1 | 49.3±2.2 | 50.5±2.6 | **54.1**±2.5 | **53.6**±4.8 |
| Concrete | **64.0**±6.1 | **63.8**±5.7 | **66.0**±7.1 | 46.2±4.9 | **55.9**±6.2 | 51.9±3.5 | 49.7±3.9 |
| QM8* | **71.3**±9.4 | **73.5**±6.8 | **65.5**±10.3 | – | **91.7**±16.9 | – | – |
| QM9* | **62.7**±3.6 | **64.9**±3.5 | **55.0**±14.4 | – | **56.8**±28.4 | – | – |

| MAD | LVD | MADSplit | CQR | DJ | DE | MCDP | PBP |
|---|---|---|---|---|---|---|---|
| Yacht | 1.90±0.48 | 1.90±0.48 | 3.55±0.85 | 10.15±0.84 | 11.25±0.81 | 10.92±0.73 | **1.80**±0.30 |
| Housing | 3.31±0.53 | 3.31±0.53 | 3.44±0.33 | 3.69±0.33 | 4.42±0.39 | 6.04±0.54 | 7.94±1.97 |
| Energy | 2.99±0.75 | 2.99±0.75 | 3.44±1.04 | 3.19±0.51 | 3.79±0.31 | 8.12±0.59 | 11.66±2.24 |
| Bike | 0.04±0.03 | 0.04±0.03 | 7.34±3.51 | 0.05±0.03 | 3.37±2.90 | 124.57±2.68 | 162.21±2.58 |
| Kin8nm | **0.07**±0.00 | **0.07**±0.00 | 0.08±0.01 | 0.09±0.01 | 0.19±0.01 | 0.18±0.00 | 0.22±0.12 |
| Concrete | 5.44±0.53 | 5.44±0.53 | 6.21±1.05 | 5.58±0.58 | 7.22±0.76 | 13.75±0.69 | 20.59±3.57 |
| QM8* | **0.01**±0.01 | **0.01**±0.01 | 0.03±0.02 | – | 3.28±5.12 | – | – |
| QM9* | **3.69**±9.09 | **3.69**±9.09 | 32.11±50.42 | – | 268.32±357.01 | – | – |

We repeat all experiments 10 times and report mean and standard deviations. For QM8 and QM9, we report the average numbers across all sub-tasks, with a breakdown on each sub-task in the Appendix.

**Results**: For **validity**, as shown in Table 3, LVD achieves marginal coverage empirically, as well as MADSplit[5] , CQR, and DJ. However, for tail coverage rate, only LVD consistently covers at or above target coverage rates. For the larger datasets, both coverage rates tend to get close to 90% for LVD. DE, MCDP, and PBP do not achieve meaningful coverage (either too high or too low). [19] also report mixed results on marginal validity with existing uncertainty quantification methods for DL. To

---

[5]It is worth noting that MADSplit, despite the theoretical guarantee, misses on the Yacht dataset, because the MAD-predictor predicts a "negative" absolute residual for some subset of the data, thus creating extremely narrow PIs, even after requiring the prediction to be positive and the "practitioner's trick" mentioned in [29].

further test for local validity, we also examine the average coverage rate conditioned on the presence of certain functional groups for the QM9 dataset (detailed results are relegated to the appendix). LVD achieves empirical validity for these groups as well, even though functional groups define a kind of similarity that is never used in the uncertainty quantification process.

For **discrimination** (Table 4), LVD is generally in the top two while maintaining the lowest MAD almost always. MADSplit has the same MAD as LVD (using the same $\hat{\mu}^{NN}$), and has a similar AUROC as LVD despite explicitly modeling MAD. Other baselines occasionally show a significant discriminative property, but usually have much higher MAD. Despite training an ensemble of models, DE incurs huge prediction errors in many datasets. As noted earlier, AUROC alone is misleading if the MAD is high: MCDP and CQR seem highly discriminative on the Bike dataset, mostly due to the high model error (epistemic uncertainty).

**Scalability:** For the largest dataset, QM9, the extra inference time of LVD vs. inference time of the original NN is 0.65 vs. 0.75 second per 1000 samples[6] on an NVIDIA 2080Ti GPU. MADSplit on the other hand takes 0.93 second overhead in the most optimized case. That said, any method that finishes within $O(1)$ multiple of the original NN model is usable in practice. For LVD, extra vs. original training time is about 1.5 vs. 0.75 second per 1000 samples, but because the $\mathbf{f}$ is already highly informative, the training of the kernel $K_{\mathbf{f}}$ finishes in very few iterations, resulting in $< 5\%$ overhead of the total training time. Note that MADSplit will take strictly $\geq 1\times$ time in total because it needs to train a second model to predict residuals. Like MADSplit, CQR needs to train at least one quantile predictor[7], but it needs to train a new predictor for every $\alpha$, which is a huge cost.

## 5   Conclusion

This paper introduces LVD, the first locally valid and discriminative PI estimator for DL, which is also scalable and post-hoc. Because LVD is both valid and discriminative, it can provide actionable uncertainty information for the real world application of DL regression models. Moreover, it is easy to apply LVD to almost any DL model without any negative impact on the accuracy due to its post-hoc nature. Our experiments confirm that LVD generates locally valid PIs that cover subgroups of data all other methods fail to. It also exceeds or matches the performance in discriminative power while offering additional benefits in scalability and flexibility. We foresee that LVD can enable more real-world applications of DL models by providing users actionable uncertainty information.

### Acknowledgments

This work is in part supported by National Science Foundation award SCH-2014438, IIS-1418511, CCF-1533768, IIS-2034479, the National Institute of Health award NIH R01 1R01NS107291-01 and R56HL138415. The authors are also thankful to Andrew Gordon Wilson and three anonymous reviewers for their comments to help improve this work.

---

[6]We use the full $\mathcal{S}_{\text{conformal}}$ for PI construction, as the inference time is short enough without sampling.
[7]That is, if one is willing to have mean estimate outside the PI occasionally and consider CQR post-hoc.

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
