# OpenReview forum: "Locally Valid and Discriminative Prediction Intervals for Deep Learning Models"
_NeurIPS.cc/2021/Conference — NeurIPS 2021 Poster_

### Official Review · Reviewer_nQpG · 2021-07-16

**Rating:** 3
**Confidence:** 4

**Summary:**

The paper focuses on quantifying predictive uncertainty through conformal prediction framework. While classic conformal guarantees are marginal (over calibration and test data), the authors aim to take a step forward towards conditional validity, which is itself an important question as typically a user prefers an uncertainty set that is conditional on a given input.

**Limitations And Societal Impact:**

The question of uncertainty quantification has a huge impact on the applicability of ML models in real-world settings and thus is receiving a lot of attention in the research community these days. Unfortunately, the current work does not clearly explain the limitations of the proposed procedure, and the comparison with alternative frameworks is questionable.

**Main Review:**

A question of conditional validity for the conformal framework has received a lot of attention over the past years. On the theory and clarity side, the paper contains significant flaws. I believe that even formalizing the problem, e.g., validity in section 2.2 requires revisiting. The writing can also be improved substantially as the paper contains a large number of typos. Examples include (there are many more):
1. Links to the equations are entered manually instead of using hyperlinks (line 87 --> "E.q. 2", lines 101 and 103).
2. Line 112 --> missing fullstop.
3. I believe that the citation given on line 93 is incorrect.
4. Equations are parts of the sentences (punctuation is needed).
5. In general, it is important to differentiate between confidence and prediction intervals not to introduce confusion. Authors focus on the latter while referring to the former.

Finally, regarding the empirical evaluation, I am not convinced that presented simulations illustrate the conditional validity of the proposed procedure. Conditional validity is easy to study at least through simulated examples (see, for example, Romano et al. "Classification with Valid and Adaptive Coverage", which is in the classification setting but might point out relevant ideas).

Unfortunately, the current work seems not to be ready to be accepted to the conference.

**Time Spent Reviewing:**

3

---

> ### Author Response · Authors · 2021-08-10
> **Response to Reviewer nQpG**
>
> ### Comment 1
> Missing hyperlink for equations (E.g. L87, Eq.(2))
>
> ### Response
> This comment is incorrect. We could not find an example of this issue, and the example pointed out by the reviewer at L87 has the correct hyperlink.
> You could try to open the pdf in Adobe Acrobat.
>
> ### Comment 2 and 4
> Missing Punctuation
>
> ### Response
> We tried our best to be grammatically correct, and we will fix the missing punctuation marks.
>
> ### Comment 3
> Incorrect Citation at L93
>
> ### Response
> L93 is essentially a direct quote from [1] (See (12)-(15) in [1]).
> The reviewer should be more specific in pointing out what's incorrect about this citation, and we will gratefully correct the error.
>
> ### Comment 5
> Confidence Interval vs Prediction Interval
>
> ### Response
> We agree that we should make the distinction between confidence interval and prediction interval (and we do mean the latter). This is a misuse of the term we carried over from many recent DL UQ papers that we extensively studied, which we will fix.
>
> ### Comment 6
> Using simulation to illustrate conditional validity
>
> ### Response
> Thank you for the suggestion.
> We intentionally avoid using simulation data to verify conditional/local validity because that involves defining what is "similar" ourselves, which will surely bear some bias.
> In other words, if we have the "similarity" definition in mind when we devise our method, our method will likely perform well, so any results will not be very convincing.
> Therefore, we used the simulation data only to qualitatively examine the behaviors of different UQ methods, but chose TCR on diverse real-world datasets to demonstrate approximate conditional validity.
>
>
> References mentioned in this response:
>
> [1] Ryan J. Tibshirani, Rina Foygel Barber, Emmanuel J. Candes, and Aaditya Ramdas. Conformal
> prediction under covariate shift, 2020.

---

> > ### Comment · Reviewer_nQpG · 2021-08-31
> > **follow-up**
> >
> > I would like to thank the authors for a detailed response. Some follow-up comments:
> > 1. "Missing hyperlink for equations": keeping one idea in mind, I described the other. My intention was to point out typos here (L87: "E.q." should be replaced with "Eq.") and inconsistency between lines 101 and 103 in order to improve visual presentation. of the paper.
> > 2. The goal of using simulated data is to further support the point but not to replace simulations on real data: for example, the true conditional distribution of the response could be known.
> >
> > Some of the other issues I observed were already pointed out by other reviewers and resolved, but I am also not convinced that the definition of local coverage in Eq. 4 is correct in the current version of the work (after consulting with cited works). While the conditional part is correct (i.e. conditioning is indeed on $X_{N+1}$), it seems like a data point from the localized distribution should be on the left.

---

> > > ### Author Response · Authors · 2021-09-01
> > > **Updates**
> > >
> > > 1. "E.q." will be replaced with "Eq.".
> > > The spacing between L101 and 103 comes from the NeurIPS template (assuming the "inconsistency" refers to this).
> > >
> > > 2. (Assuming by "simulations on real data", you mean "experiments on real data".)
> > > We would extend our simulation data experiment and add a "coverage vs $x$" plot to the Appendix.
> > > However, we'd like to elaborate on why we opposed using simulation data.
> > > First, LVD is only "approximately" conditionally valid, so it does not make sense to create any "adversarial" simulation study.
> > > Then, since the focus is more on the "approximate" aspect, it will be very hard to come up with a simulation data without bias.
> > > For example, if L2 distance is used as the similarity measure, not choosing the "adversarial route", we should require closer (as measured by L2 distance) data to generate similar responses and noises.
> > > But in this case we should simply use L2 distance instead of training a kernel.
> > > The same argument applies to any distance metric we have in mind when creating such simulation.
> > >
> > >
> > > 3. The current version of Eq. (4) is correct, and it should not be a data point from the localized distribution on the left.
> > > This confusion might come from Eq.(3), which is a different definition of "local" validity than Eq.(4).
> > > Such difference however fades away in the asymptotic case.

---

### Official Review · Reviewer_nv6c · 2021-07-16

**Rating:** 6
**Confidence:** 3

**Summary:**

The authors propose a new method for locally valid (e.g. marginal coverage guarantees) and discriminative prediction* intervals by connecting two ideas: kernel regression and conformal inference. They evaluate their method on a suite of small to medium scale regression tasks and find that the proposed approach is locally valid


*note that the authors refer to these intervals as _confidence_ intervals which I believe to be an error. CIs seek to cover a population parameter, which for regression would be the true, unknown, and underlying mean function (which can only be assessed via a simulation). What the authors propose are _prediction_ intervals which seek to cover draws from this distribution and are inclusive of distributional noise, c.f. this reference for more:

https://www.stat.cmu.edu/~ryantibs/papers/conformal.pdf

**Limitations And Societal Impact:**

Yes

**Main Review:**

### Main comments:
I like the combination of kernel regression and split conformal inference. Overall, I found the method compelling and backed by a solid theoretical argument, though I found the experimental results somewhat limited, both in terms of the interval construct methods that were evaluated, deep learning models, and datasets. There also is a semantic confusion (see above) around confidence vs prediction intervals that warrants addressing.

Additionally, the method relies on the assumption that neural nets (i.e. the embedding function) learns a mapping such that "similar" inputs end up mapping near to each other in this new space. I believe there is a lot of evidence in the literature for this assumption but it is worth making this assumption explicit.


### Detailed comments:

__Originality__: The work is original and combines two ideas to produce a novel uncertainty quantification procedure for deep learning models. Note, however, very similar ideas have been explore for non-deep learning models in the statistical literature, e.g.:

https://www.tandfonline.com/doi/abs/10.1080/01621459.2012.751873

__Quality__: The experiments are of high quality and I offer some suggestions for strengthening them below:

- The investigation into conditional coverage is limited to tail coverage and, given it is listed as one of the central contributions of the paper, I would like to see more empirical evaluations where other, non-tail subsamples are evaluated.

- Given the close relationship between kernel methods and Gaussian Processes, it would be nice to include a GP in this comparison. Given the size of some of the datasets, I appreciate GPs could be expensive but it would be nice to see the comparison of how a 90% posterior interval from a GP performs.

- Evaluations on other kinds of architectures (e.g. CNNs) would also greatly strengthen the results.

- Prior evaluations of marginal coverage properties of Bayesian and non-Bayesian deep learning models should be acknowledged:

https://arxiv.org/abs/2010.03039

__Clarity__: The work is clearly presented and I found text and experiments easy to follow.

__Significance__: The work addresses an important topic and will be of interest to applied and theoretical researchers.

**Time Spent Reviewing:**

2 hours

---

> ### Author Response · Authors · 2021-08-10
> **Response to Reviewer nv6c**
>
> ## Comment 1:
> Assumption that NN learns to map similar data closer together
>
> ## Response:
> With finite sample size, the assumption that NNs learn to embed similar inputs near each other is indeed necessary and implicit. It should be noted that as the number of samples increase, this assumption is not as critical, because of the asymptotic conditional validity like in [1] and [4]. We will add this clarification in the revised version.
>
> ## Comment 2:
> Similar work ([3]) in non-DL settings
>
> ## Response:
> Thanks for pointing out [3], which we will add to the references. There are indeed significant efforts in statistics tackling this issue, including [4] (by the same authors of [3]). Our work focuses more on the practical application of statistical theory to the DL setting, and we are fortunate to be able to build on ideas from the rich literature in conformal prediction.
>
>
>
> ## Comment 3:
> Suggestions on the Experiments
>
> ## Response:
> Regarding the experiment section, we greatly appreciate your insightful suggestions. Although we cannot conduct all the experiment by the author response deadline, we will add additional experiments for the camera-ready version:
>
> ### Comment 3.1:
>
> more evaluations on non-tail subsamples of "similar data"
>
> ### Response:
>
> The reason why we focus on tail coverage is that we think the two tails are "groups" that are less bias-prone.
> If we hand-pick some "similar" data basing on $X$ to show that LVD achieves better coverage within that group, we worried that our bias will make the result less convincing.
> We will however explore different "groups" of molecules in the QM datasets, and add additional evaluations in the appendix
>
> ### Comment 3.2:
>
> Add Gaussian Process (GP) as a benchmark
>
> ### Response:
> We did not include it because our focus is on the DL applications. This is also why we picked MCDP, which is very popular in DL, but is also similar to GP in its behavior.
> We ran the experiments using GP on the non-QM datasets, and the results are shown below.
> We used the implementation in scikit-learn, with RBF and WhiteKernel for prediction intervals.
> For MCR and TCR, numbers not significantly lower than 90 are marked bold.
> For AUROC, numbers significantly higher than 50 are marked bold.
> From the results GP sometimes covers marginally, but never covers the tails, and the PI width does not seem to correlate with the actual error very well as well. MADs for GP are also generally quite high. In general GP and MCDP's behaviors are very similar.
>
> | GP          | MCR         |  TCR   |  AUROC   | MAD     |
> | ----------- | ----------- | ----------- |----------- |----------- |
> | Yacht | **87.7**$\pm$4.2 |50.8$\pm$9.7|50.0$\pm$0.0|9.83$\pm$1.36|
> | Housing | **92.7**$\pm$3.7 |65.2$\pm$17.2|47.3$\pm$4.4|7.82$\pm$0.82|
> | Energy | 88.1$\pm$1.5 |48.7$\pm$1.0|47.1$\pm$5.2|8.95$\pm$0.31|
> | Bike        | 89.1$\pm$0.6| 49.9$\pm$0.0 |40.7$\pm$1.6|167.12$\pm$2.53|
> | Kin8nm      | **89.9**$\pm$0.7| 85.7$\pm$1.0|**56.4**$\pm$0.9|0.06$\pm$0.00|
> | Concrete | **89.7**$\pm$2.7 |53.9$\pm$7.8|**56.1**$\pm$3.9|19.31$\pm$0.80|
>
> ### Comment 3.3:
>
> Experiments on other NN (like CNN)
>
> ### Response:
>
> Given the popularity of CNN, it would indeed be good to include it. We chose the MPNN+molecule property regression because this is a popular regression task using DL, whereas CNN is more often used in classification.
> We plan to add experiments on the pose regression task, which would permit a CNN model.
>
>
> ### Comment 3.4:
>
> Additional reference [5]
>
> ### Response:
>
> Thanks for the suggestion. We will acknowledge [5] in Section 4.2 (from L265).
>
> References mentioned in this response:
>
> [1] Leying Guan. Conformal prediction with localization, 2020
>
> [2] Ryan J. Tibshirani, Rina Foygel Barber, Emmanuel J. Candes, and Aaditya Ramdas. Conformal
> prediction under covariate shift, 2020.
>
> [3] https://www.tandfonline.com/doi/abs/10.1080/01621459.2012.751873
>
> [4] Jing Lei and LarryWasserman. Distribution-free prediction bands for non-parametric regression.
> Journal of the Royal Statistical Society: Series B (Statistical Methodology), 76(1):71–96, 2014.
>
> [5] https://arxiv.org/abs/2010.03039

---

### Official Review · Reviewer_kPFV · 2021-07-17

**Rating:** 7
**Confidence:** 5

**Summary:**

This paper considers the problem of providing prediction intervals in the deep learning setting, leveraging the recent work "Conformal prediction with localization" by Guan [2019]. The prediction intervals provided by this framework are simultaneously marginally valid, as well as conditionally valid in a certain sense, unlike other conformal methods.

**Ethics Review Area:**

["Research Integrity Issues (e.g., plagiarism)"]

**Limitations And Societal Impact:**

I don't think the method has any serious limitations more than other conformal methods. The authors briefly mention this in Sec 2.2.

**Main Review:**

The empirical study done by the paper is extensive and has not been done before. However, the paper **significantly** overstates the novelty in terms of theory and methodology. The main algorithm and theoretical results heavily rely on the work of Guan [2019, 2021], as can be verified by looking at the proof in Appendix A. In particular, the conditional-style coverage result (equation (4) or Theorem 3.1) is not novel. The way the paper is written strongly indicates otherwise. I expand further on this point in the 'Writing' section below. **I strongly suggest that the authors transparently discuss the relationship to Guan's work, starting with a citation in the abstract.** I request the authors to correct me if I am missing some important theory developed in this paper which was not present in Guan's work, in which case I will reconsider my assessment.

_Strong empirics_:

As far as I know Guan's proposed method of localization, although having very nice theoretical properties, has not been empirically validated in an extensive study. This paper provides such a study for real world datasets for deep learning models; Section 4 confirms that the localization framework leads to discriminative prediction intervals that are marginally valid as well as have better conditional performance. The authors have considered many standard real world datasets, compared their approach to multiple other UQ methods (conformal and non-conformal), and used multiple meaningful evaluation metrics. They comment also on the time required for training and prediction of their method vs other methods.

_Methodology_:

This paper discusses learning an adaptive kernel based on the Nadaraya-Watson estimator. Guan's paper considered the localization function or kernel as fixed; thus learning the kernel using data is a new methodological contribution.

Once the kernel is learnt, from what I can tell the conformalization technique is exactly the same as Guan's paper, and so are the theoretical guarantees. Is there further development regarding the conformal methodology or theory itself which I have missed?

_Writing_:

The writing is clear and easy to follow. However, I feel that there are two significant issues which make the writing unscholarly to the extent that it has affected my overall score.

- **The paper seems to significantly downplay the dependence of their methodology and theory on Guan's paper.** The first mention of Guan's paper appear in passing in Section 2.2, which I found very strange. Lines 54-56: "... most conformal methods are only marginally valid [24, 37, 19, 4]. We however seek to construct a CI conditioning on the input." Is there a reason for not discussing the work of Guan [12] in this context? To a reader, it appears that this paper proposes a new solution to the conditional validity problem, whereas in fact the solution used is exactly the one proposed by Guan. Similarly lines 58-66 incorrectly suggest that the method proposed by the authors is new and this problem has not been solved before. Line 286 "This paper introduces LVD, the first locally valid and discriminative CI estimator for DL" also suggests that the conformal methodology is new.

- The standard terminology for intervals that cover the output Y_{N+1} in _all_ of the conformal literature is ‘prediction intervals’, whereas ‘confidence intervals’ would attempt to cover some parameter of the distribution of Y, such as E[Y_{N+1} \mid X_{N+1}]. It seems that the authors have made a deliberate choice to use the term confidence intervals. Could the authors clarify the reason for this choice? If there is a strong reason, I suggest the authors discuss this in the paper. Else, I suggest to use 'prediction intervals' as is done in other papers on conformal.

_Additional minor comments on writing and related works_:
- The abstract is quite long, and contains many finer details and terminologies. These can be postponed to the introduction or discussion sections. I suggest streamlining the abstract and focusing on the 'big picture' to reach a wider audience.
- Line 39: “Post-hoc methods are ideal because ...”. While this is common, I believe it is not a necessary property of post-hoc methods.
- Line 50: “Such methods usually use the prediction errors from a hold-out set to ...” Such ‘split’ conformal methods are currently popular due to their efficiency and validity despite their simplicity. However the statement of the authors is a misrepresentation of the well-developed body of conformal prediction where many methods that use all of the data exist: full conformal in Vovk’s 2005 book, cross-conformal [2], jackknife+ [3], and out-of-bag conformal methods [4].
- The paper [1] is highly relevant and should be cited somewhere.
- Eq. (4): Should X have a \tilde above it?
- Line 112: Missing period ‘.’ after closing parenthesis.
- Line 146: The definition of R_i may be somewhat abrupt for readers unfamiliar with conformal prediction

[1] https://arxiv.org/abs/1905.10634

[2] http://proceedings.mlr.press/v91/vovk18a.html

[3] https://arxiv.org/abs/1905.02928

[4] https://arxiv.org/abs/1910.10562

**Needs Ethics Review:**

Yes

**Time Spent Reviewing:**

8

---

> ### Author Response · Authors · 2021-08-10
> **Response to Reviewer kPFV**
>
>
> Thank you for your pointed comments as well as the positive assessment of the extensive experimental work that the paper (and the accompanying appendix) reports. We appreciate your feedback, which we will readily incorporate to strengthen the paper.
>
> Below we address your comments in order, beginning with the two major points.
>
>
> ## Comment 1
> Framing of paper and relation to work by Guan [1].
>
> ## Response
>
> ### What's the problem we are solving
> We first stress that our work focuses on uncertainty quantification (UQ) *in deep learning (DL)*.
> There were significant recent efforts to design good UQ methods for DL, but many have significant limitations or issues.
> For instance, we analyzed [3] in detail in Section 4 and Appendix C, showing that its claims of validity and discriminability are questionable.
>
> ### Differences
> You mention that [1] already solves our task, but in fact, it proposes a very high-level idea that does not just work in our setting.
> We point out several crucial differences between LVD and [1], which are discussed in Sections 3.1 and 3.3.
> Examples include:
> - the use of a flexible kernel instead of a fixed one (unlike in [1] and [2]).
> - learning a metric on the embeddings, as training the kernel end-to-end from scratch is not feasible
> - a suitable smoothness regularization, which is key to avoid unreasonably wide intervals for some datasets (the detailed ablation is reported in the appendix).
>
>
> ### We did not downplay [1]
> It was never our intention to downplay the connection between our work and [1].
> When we present the theoretical analysis, we state right before the theorem that this is an adaptation of *"Theorem 5.1"* from [1], so as to avoid confusion that this is *our* theory.
> [1] is also explicitly referenced over 10 times in the explication of the proofs, as well as when we define local validity, so we do not think this could be considered plagiarism.
> Since our target audience is the DL community, when we claim "ours is the first method...", we tried our best to limit the scope to black-box and expensive models like DNN in places like L286 (as you highlighted).
> We also kept Section 3.4 very short, because we consider it a minor adaptation of established theory, and not the contribution of our work.
> Also, the idea of local validity and using kernels is in several earlier works we cite as well (E.g. [2] and [4]), and we did not consider LVD as an application *completely* basing on [1], but an example of combining these conformal methods and ideas with DL practically.
>
>
> ### Why not cite [1] earlier
> We present definitions in the current order in Section 2.2 because we considered this ordering more natural (especially with [2] preceding [1]) for unfamiliar readers.
> We tried your suggestion about mentioning [1] in Section 1 before but found its preceding of the definitions (Section 2.2) out of place.
> **We will however make suitable changes to address this concern** (see "Action" a&c below).
>
>
> ### Actions
> Finally, we will make the **following updates** to the text to further clarify the relation:
>
> a. Add a few lines, in the beginning, mentioning [1], [2], and [4] as our major inspirations (particularly [1]), that we adapt to the DL setting (e.g. in L60)) and how the experimental evaluation will be valuable.
>
> b. Clarify at L11 that the "recent advances" are [1] and [2].
>
> c. Add a sentence at L55 mentioning [1] and [2] as "approximate conditional" but defer the definitions to Section 2
>
> d. Add a reference to [1] in the abstract.
>
> ## Comment 2
> Confidence Interval vs Prediction Interval
>
> ## Response
> We acknowledge this distinction between confidence interval and prediction interval, which is indeed quite basic in statistics (most of the statistics papers we cite use this terminology, as expected). However, you are right, this was indeed a deliberate choice, albeit a weak one and not without consternation. Many of the papers we extensively studied in the DL/ML domain used them interchangeably (like [3], [6], and [7]).
> As noted earlier, our focus is UQ in DL, and we tried to write in the language of those works (especially [3], which we hoped to demystify). To avoid perpetuating this erroneous usage, **we will change it to "prediction interval"**.
>
> ## Minor comments
>
> - *Abstract too long*: Thank you for the suggestion and we will try to shorten it.
>
> - *Post-hoc Method might interfere with the base model*: This is true but very rare. We will add "usually" to make it more rigorous.
>
> - *Different splitting methods (L50)*: We were constrained by space and did not expand on more conformal methods, but will rephrase this sentence to avoid confusion for readers new to conformal prediction.
>
> - *Reference [5]*: We were not aware of [5] but it is an interesting alternative and we will discuss it in Section 1.
>
> - *$\tilde{X}$ in Eq (4)*: We followed the definition in [1], and there should not be a tilde here.
>
> - *Missing period at L112*: We will fix this.
>
> - *L146 definition of residuals*: We will add a short comment below the formula.
>
> References mentioned in this response:
>
> [1] Leying Guan. Conformal prediction with localization, 2020
>
> [2] Ryan J. Tibshirani, Rina Foygel Barber, Emmanuel J. Candes, and Aaditya Ramdas. Conformal
> prediction under covariate shift, 2020.
>
> [3] Ahmed Alaa and Mihaela van der Schaar. Discriminative Jackknife: Quantifying Uncertainty
> in Deep Learning via Higher-Order Influence Functions. ICML 2020 119: 165-174
>
> [4] Jing Lei and Larry Wasserman. Distribution-free prediction bands for non-parametric regression.
> Journal of the Royal Statistical Society: Series B (Statistical Methodology), 76(1):71–96, 2014.
>
> [5] https://arxiv.org/abs/1905.10634
>
> [6] Ahmed Alaa and Mihaela van der Schaar. Frequentist Uncertainty in Recurrent Neural Networks via Blockwise Influence Functions. ICML 2020, 119: 175-190
>
> [7] Volodymyr Kuleshov, Nathan Fenner, and Stefano Ermon. Accurate Uncertainties for Deep Learning Using Calibrated Regression. ICML 2018, 80: 2801-2809

---

> > ### Comment · Reviewer_kPFV · 2021-08-24
> > **Thank you for an elaborate response**
> >
> > The suggested _Actions_ would definitely improve the contextualization of this paper with that of Guan's. Regarding 'prediction intervals', perhaps you can add a footnote or brief remark on how they are different from confidence intervals (for unfamiliar readers).
> >
> > I have now increased my score to 7, assuming that the proposed changes will be made. The paper successfully adds strong empirical credibility to Guan's elegant theoretical result.

---

> > > ### Author Response · Authors · 2021-08-24
> > > **Updates**
> > >
> > > Thank you for your comment, as well as your suggestions to improve the paper. Propagating the suggested changes is already underway. We will also add the additional benchmarks suggested by reviewer nv6c using GP (for which we shared results on some datasets) and including a CNN-based benchmark in the appendix.

---

### Official Review · Reviewer_bNhT · 2021-07-17

**Rating:** 7
**Confidence:** 3

**Summary:**

The authors the Locally Valid and Discriminative (LVD) method to construct confidence intervals. The idea has three steps. (1) First is to take the embeddings from a neural network and train a kernel regression model on the task, learning a Gaussian kernel $K$. (2) Second is to, on a separate held-out "conformal" set, compute the residuals between the model and ground truth labels. (3) Third is to, at inference time, compute the empirical desired $(1-\alpha)$ interval on the "conformal" set residuals, where each residual is re-weighted according to $K(x', x)$ where $x'$ is the query of interest. LVD has the appealing property that it is compatible with any (potentially pre-trained) deep learning model, with no further further training necessary.

The authors prove that under appropriate assumptions LVD achieves *local coverage*, i.e. coverage not only marginal over the entire feature space, but at a particular query $x$. Empirically they demonstrate that LVD improves upon popular neural network predictive uncertainty methods on the well-known UCI datasets. They evaluate in terms of Mean Absolute Deviation, Marginal Coverage Rates, and Tail Coverage Rates (which is defined as the coverage rate for the left and right 10% of the tails of labels $y$).

**Ethical Concerns:**

No ethical concerns.

**Limitations And Societal Impact:**

Yes, the authors adequately addressed limitations and potential negative societal impact.

**Main Review:**

On the whole I was quite impressed with this paper. The authors write clearly throughout. Experimental setup is reasonable, and Figure 1 in particular provides a succinct picture for their motivation.

I was left with a few questions and suggestions:

1/ How does the smoothness requirement affect the confidence interval estimates? It would be interesting to repeat the procedure of say Table 3 and 4 as an ablation study to determine its necessity.

2/ It would be helpful to explicitly define Mean Absolute Deviation (MAD) in Section 4.2.

3/ Pearce et al. ICML 2018 directly measures the Mean Prediction Interval Width instead of the MAD, as a measure of discrmination. I'd be curious to see that added as a sub-table of Table 4 (perhaps in the Appendix), to compare the two evaluation metrics.

Miscellaneous comments, typos:

1/ It looks like references 18, 19 are duplicates of each other.

2/ Line 221: "Conforamlized Quantile Regression".

--

Update: I thank the authors for their response, and for pointing me to Tables 7-10 in the Appendix in particular. After reading through their response, and the other reviewers' comments, I maintain my recommendation that the paper be accepted.

**Time Spent Reviewing:**

4

---

> ### Author Response · Authors · 2021-08-10
> **Response to Reviewer bNhT**
>
>
> We really appreciate your comments and feedback. We will of course correct typos and remove the duplicate reference. For your specific comments, please find responses below:
>
> ## Comment 1
> Impact of the smoothness requirement.
>
> ## Response
> The smoothness requirement turns out to be crucial for tighter intervals. We believe tables 7-10 in the appendix will answer your question. Tables 9 and 10 show that the smoothness requirement produces tighter intervals. Tables 7 and 8 show that it does so without sacrificing other metrics.
>
> ## Comment 2
> Explicit definition of MAD.
>
> ## Response
> Thanks for the suggestion. We will define it explicitly.
>
> ## Comment 3
> Mean Prediction Interval Width vs MAD in evaluation.
>
> ## Response
> Thank you for the suggestion. MAD measures the discriminability of the point estimate, whereas the interval width measures that of the interval and they are indeed complementary to each other.
>     We have Table 5/9/10 in the Appendix measuring mean prediction interval width for the *finite* intervals.
>     It is somewhat hard to compare this metric, however, because to satisfy the stricter local coverage property, LVD needs to output infinite intervals when there's little data in the neighborhood.
>     We can see that LVD gives the shortest intervals for many datasets (Bike, QM8, and QM9) with few infinite intervals though.

---

### Review · Ethics_Reviewer_i6fb · 2021-07-22

**Recommendation:**

In the interest of better situating their work, the authors may wish to examine how they address related work.

**Ethics Review:**

As pointed out by one of the technical reviewers, the authors have not adequately contextualized their work with respect to that of Guan. This does seem to be poor scholarship, but it does not seem to me to be an instance of plagiarism.

---

### Review · Ethics_Reviewer_rFke · 2021-08-11

**Recommendation:** N/A

**Ethics Review:**

No explicit ethical issues beyond general ML. (No experiments involving human data/judgments. No explicit unethical experimentation.)

---

### Decision · Program_Chairs · 2021-09-27

**Decision:**

Accept (Poster)

**Comment:**

Three reviewers indicated acceptance, two of them with clearly positive scores. Their main arguments in favor of this paper were good methodological contributions, convincing experiments, clarity of motivation and arguments, and a convincing rebuttal that added many details and addressed most points of criticism. On the other hand, there was one clearly negative review, but during the discussion period, the other members of the review committee (including myself)  had the impression that the authors' rebuttal addressed all these pints of criticism in a detailed and convincing way. So I recommend acceptance of this paper.